# Accelerated Aging of Epoxy Biocomposites Filled with Cellulose

**DOI:** 10.3390/ma15093256

**Published:** 2022-05-01

**Authors:** Radosław Busiak, Anna Masek, Aleksandra Węgier, Adam Rylski

**Affiliations:** 1Institute of Polymer and Dye Technology, Faculty of Chemistry, Lodz University of Technology, Stefanowskiego 16, 90-537 Lodz, Poland; radoslaw.busiak@dokt.p.lodz.pl (R.B.); aleksandra.wegier@dokt.p.lodz.pl (A.W.); 2S.Z.T.K. “TAPS”—Maciej Kowalski, ul. Borowa 4, 94-247 Lodz, Poland; 3Institute of Materials Science and Engineering, Lodz University of Technology, 90-924 Lodz, Poland; adam.rylski@p.lodz.pl

**Keywords:** epoxy resin, cellulose, biocomposites, aging

## Abstract

The presented research concerns the mechanochemical modification of a snap-cure type of epoxy resin, A.S. SET 1010, with the addition of different amounts of cellulose (0, 2, 5, 10, 15 and 20 per 100 resin), for a novel, controlled-degradation material with possible application in the production of passenger seats in rail transport. Composite samples were prepared on a hydraulic press in ac-cordance with the resin manufacturer’s recommendations, in the form of tiles with dimensions of 80 × 80 × 1 mm. The prepared samples were subjected to thermo-oxidative aging and weathering for a period of 336 h. Changes in the color and surface defects in the investigated composites were evaluated using UV-Vis spectrophotometry (Cie-Lab). The degree of degradation by changes in the chemical structure of the samples was analyzed using FTIR/ATR spectroscopy. Differential scan-ning calorimetry (DSC) and thermogravimetric analysis (TGA) tests were performed, and the sur-face energy of the samples was determined by measuring the contact angle of droplets. Tests were performed to determine changes in cellulose-filled epoxy resin composites after thermo-oxidative aging and weathering. It was found out that the addition of cellulose did not inflict sufficient changes to the properties within tested parameters. In the tested case, cellulose acted as a natural active biofiller. Our research is in line with the widespread pursuit of pro-ecological solutions in industry and the creation of materials with a positive impact on the natural environment.

## 1. Introduction to the Development of Materials Used in Rail Transport

From the 19th century to the present day, rail transport has played a key role in the transportation of goods and passengers. Since then, rail has evolved from steel and wood machines powered by steam boilers to the ultra-light composite energy-saving cars used in high-speed trains [1,2]. Today, railway transport is considered to be the most environmentally friendly way of transporting goods and people, where the indicator of this is the very low rate of carbon dioxide generated into the atmosphere per passenger kilometer (in 2014—28.39 g/pkm [3]) [4,5,6]. Modern rail transport requires the use of many different materials to achieve one common goal, which is to reduce emissions into the environment [7,8,9,10,11,12]. To protect the environment, the rail industry is taking appropriate steps to reduce emissions of noise and exhaust fumes from internal combustion engines, cut energy demand, conserve the materials required to produce train cars, and manufacture train cars that are easier to dispose of and recycle at end-of-life [7,13]. To achieve these goals, the railway industry is continuously looking for new materials, technologies and solutions [7,14]. One of the essential elements in a train are railway seats. Current trends aim to significantly reduce the weight of the seats by using lightweight and durable materials. Metal parts, despite their excellent mechanical properties, are characterized by high weight and price. Polymer composites can be a good replacement for selected metal parts of a railway seat. Numerous requirements are imposed on materials used in railway applications, such as mechanical strength, resistance to scratching and flame retardance. The industry has shown great interest in epoxy resins due to their excellent properties: chemical and dimensional stability, high specific stiffness, fatigue strength, scratch and electrical resistance, simple manufacturing, and low price [15,16].

Epoxy resins have become synonymous with durability, particularly in applications such as coatings, adhesives, and composite materials [17,18]. By convention, epoxy resins are referred to as both the uncross-linked form and the cured form. These compounds belong to a family of monomers and oligomers that can be used to create thermosetting plastics with high physical and chemical resistance, adhesion to many materials, low shrinkage after curing, high impact strength, elasticity, and excellent electric current insulating properties [17,19,20,21]. A special feature of all epoxy resins is the presence of a reactive epoxy (oxirane) ring, which is responsible for the resin’s ability to crosslink [19,22,23]. Moreover, epoxy rings show affinity to both nucleophilic and electrophilic groups and molecules. This allows the use of substances of different nature, allowing for crosslinking of resins via different pathways and mechanisms [23]. Despite the benefits of a cross-linked structure, it is difficult to recycle—the resin cannot be reshaped by heating [24]. That is why it is essential to modify the composite with natural, degradable additives.

The resin used in the experiment belongs to the category of snap-cure type resins [25]. Resins of this system are characterized by the fact that they are in a powder form at room temperature. Before using the resin, the powder should be dissolved at elevated temperature. These resins crosslink when exposed to high temperature (above 100 °C) within a few minutes to a maximum of a few hours [26,27].

Due to rising demand for environmentally friendly materials, industry started focusing on the use of natural, fully degradable fibers such as cellulose to improve composite properties and their biodegradability in order to ensure as small an impact on the natural environment as possible while following the rules of sustainable development [28,29,30,31,32].

The susceptibility of the composite to degradation can be increased by adding substances of natural origin to the polymer matrix. Polysaccharides such as cellulose dispersed in polymer matrix are mentioned in several studies. The potential of cellulose as an additive to epoxy resins is due to its biodegradability, renewability, excellent mechanical properties, and low cost. However, the dispersion of cellulose into a resin matrix is challenging because of its hydrophilic nature, high specific area, and surface energy. The solution to this issue is chemical modification, among which the methods of mercerization, silanization, oxidation and esterification are known [33,34]. This article focuses on the use of cellulose powder dispersed in a volume of thermosetting epoxy resin as a possible material for reinforced composites used in passenger railcar seat production. This is an innovative solution for the railway industry, which aims to create a safe material for passenger transport as well as an environmentally friendly one that simplifies the recycling of decommissioned railcars. The main objective of this research was to design polymer biocomposites with controlled degradability in the natural environment. Innovatively, cellulose fibers was used to increase the susceptibility of synthetic resin to degradation by natural environmental factors. The presented research is part of a larger project aimed at transforming synthetic composites used in the railway industry into biodegradable and renewable native composites.

## 2. Materials and Methods Used in the Experiment

### 2.1. Preparation of Samples

Samples were prepared from one component epoxy resin, A.S. SET 1010 (provided by S.Z.T.K. “TAPS”—Maciej Kowalski, Borowa 4, 94-247 Lodz, Poland), filled with 0, 2, 5, 10, 15 and 20 phr of dried, unmodified ARBOCEL^®^ UFC 100 by JRS Rettenmaier—a microcrystalline, ultra-fine and highly pure cellulose powder. The cellulose powder was dried in a Binder dryer at 100 °C for 24 h. Samples were prepared on a hydraulic press in the form of tiles with dimensions of 80 × 80 × 1 mm. Tiles were pressed for 4 min at 70 °C in order to plasticize the resin and cured for 6 min at 120 °C. Both stages of the procedure were carried out at pressure of 120 bar. Resin and cellulose used to prepare samples were mixed in small portions using lab cups and spoons. The process of preparing the samples is shown in Figure 1.

All prepared samples were split into three batches and aged separately. Thermo-oxidative aging was carried out in a Binder dryer at 100 °C for 336 h. The weathering process was carried out using an Atlas Solar SC 340 aging chamber for 336 h. The weathering process simulated day–night cycles by subjecting samples to fluctuating temperature (in cycles of 4 h at 10 °C and 4 h at 70 °C), UV, VIS and IR radiation and air humidity of 80%. The third batch was the control group, which was not subjected to any aging factor.

### 2.2. Methods

Determination of color change in the CIE-Lab coordination system was performed on a Konica Minolta Spectrophotometer cm-3600d. The color change test was performed against a white background. The reference sample for the test was a sample that had not undergone any type of aging. The determination of the color change was performed according to PN-EN ISO 105-J01 norm.

The determination of the surface energy was performed on a dataphysics OCA15EC goniometer using the Owens–Wendt–Eabel–Kealble method by measuring the contact angles of droplets on the surfaces of the tested samples. The study was conducted using three measuring liquids: water, ethylene glycol and diiodomethane. Wetting angle measurements were performed on all of the tested samples to compare the surface energies of all of the composites and to compare the influence of different types of aging on the surfaces of the composites. The contact angle measurement was repeated 6 times for each liquid on each of the tested samples.

In order to study the changes occurring on the surfaces of the samples, infrared spectra were taken on a Thermo Nicolet 6700 FT-IR apparatus with Smart ITx attachment for ATR testing (using diamond as a crystal). A background measurement consisting of 64 scans with a background gain of 4.0 at spatial resolution of 4 cm^−1^ with a laser at 15,798 cm^−1^ was carried out prior to the examination of the samples. The spectra obtained were presented as a function of absorbance.

The TGA test was carried out using a Mettler Toledo TGA/DSC1 apparatus for selected unaged samples in an air environment with a flow rate of 60 mL/min, in the temperature range 25–900 °C with a constant temperature rise of 20 °C/min. Ceramic crucibles (made of aluminum oxide) with a capacity of 70 μL containing approximately 5 mg of the test sample were used for the measurement. The blank standards used to calibrate the TGA/DSC1 apparatus were zinc and indium.

The temperature ranges of the phase transitions occurring in the composites studied were determined using differential scanning calorimetry (DSC). The study was carried out using a Mettler Toledo DSC1 apparatus. For the measurement, 5 mg of shredded material from the center of the test sample was used. Aluminum crucibles with a capacity of 40 μL were used in the test. The DSC measurement was carried out in the temperature range from 25–250 °C at a heating rate of 10 °C/min. The test was conducted in an inert gas atmosphere (argon), the flow rate of which was set at 60 mL/min. The standards against which the measuring apparatus was calibrated were n-octane and indium. The heat of indium was determined according to the heat of fusion of indium. Which was 28.45 J/mg. The results of the study were energy difference curves as a function of temperature, which were obtained using the STARe program. Based on the obtained curves, it was possible to determine the temperatures of the transformations occurring inside the investigated samples as well as their thermal effects.

The particle dispersion test was performed using an OPTA-TEC LAB-40 optical microscope and Capture V2.0 software. The test was performed by counting cellulose particles observed in the sample. The magnification of the microscope used to observe the samples was 100 times. The sample area covered by the microscope observation was 12,538.78 µm^2^. Observations were made at three different locations on each sample.

## 3. Results

### 3.1. The Impact of the Aging Process on Color Change of Tested Samples

It was observed that the reference samples (not containing cellulose) subjected to thermo-oxidative aging changed their color slightly in comparison with the preaged state—the value of the differential (which is a measure of the overall change in color) dE ab = 1.16.

Increasing the amount of cellulose in the tested samples made them more prone to aging processes at high temperature, which was marked by a rising hue change. The largest visual changes after thermo-oxidative aging were observed for sample with addition of 20 phr of cellulose (differential dE ab = 10.18).

On the other hand, there was a decreasing trend of hue change with increasing cellulose content that could be observed. The sample showing the least change in color after the weathering process in the sample containing 5 phr cellulose (differential dE ab = 24.8).

Analyzing the results obtained, it could be concluded that the aging of A.S. SET 1010 epoxy resin and its composites was influenced much more by solar radiation than by the effect of elevated temperature alone. Solar radiation is responsible for initiating free radical degradation processes in structures of epoxy resin and causes the formation of excited states in the polymer chain, thus creating free radicals in the polymer matrix. The collected results are presented in Table 1 and Figure 2. 

### 3.2. Analysis of the Effects of the Aging Process on Free Surface Energy of Resin Biocomposites

On the basis of the obtained results, it was observed that all tested samples displayed hydrophobic properties (the values of contact angles for water droplets were higher than 90°). Moreover, the content of dispersed cellulose in the whole volume of the resin had no visible effect on the surface properties of the material. The type of aging to which the samples were subjected also had no noticeable effect on the changes in the surface properties of the resins and their composites. The samples subjected to simulated aging processes continued to exhibit hydrophobic properties, i.e., the observed wetting angles for water continued to be greater than 90°, as shown in Figure 3.

Analyzing the surface energies obtained from the wetting angle measurements of water, ethylene glycol and diiodomethane droplets, the total surface energies did not change drastically regardless of the cellulose content of the sample or the type of aging process the sample underwent. However, the sample made of A.S. SET 1010 resin without added cellulose and subjected to climatic aging showed the highest total surface energy at 33.27 mN/m as shown in Figure 4. The same sample also showed the lowest polar component (sample with the highest hydrophobic character). The values of the total surface energies are shown in Figure 5.

After further analysis, it was observed that thermo-oxidative aging generally caused an increase in the polar component of the surface energy. The only exception was the non-aged sample containing 10 phr of cellulose. The polar component of free surface energy for this sample was 1.89 mN/m, which was the highest collected result. The consequence of this type of aging may be a higher hydrophilicity of the material, which in natural conditions may facilitate the acceleration of aging processes in the material. 

The values of the polar component of the surface energies are shown in Figure 5. The tested conditions appeared to have no significant effect on the changes in the surface properties of the tested materials, which was evidenced by the results presented in Figure 4. It was observed that the values of the surface energies did not differ significantly, except for the previously mentioned sample made of resin without cellulose addition (33.27 mN/m) and the sample made of resin with 20 phr cellulose addition, subjected to weathering (28.41 mN/m).

### 3.3. Discussion of Obtained DSC Curves

The curves obtained by DSC analysis allowed us to determine the glass transition temperature of the studied composites. In the examined temperature range, only two transformations occurring in the polymer composites were visible. The first endothermic peak (mid peak) corresponded to the glass transition, while the second exothermic peak corresponded to the crosslinking processes occurring in the temperature range from 100 to 180 °C. The sample containing 15 phr of cellulose showed the greatest difference in glass transition temperature (Tg = 68.32 °C). It can be assumed that this sample was much better crosslinked at the forming stage than the other samples because the exothermic peak on the DSC graph for this sample was much less intense and occurred in a much smaller temperature range (120–160 °C). The glass transition temperatures obtained for the samples studied are shown in Table 2. Appropriate DSC graphs are shown below in Figure 6.

### 3.4. Discussion and Overview of FTIR/ATR Spectra

In all spectra, peaks were observed in the following bands: 3000–2800 cm^−1^ (triplet, -CH stretching highlighted in yellow), 1606 cm^−1^ (C=C stretching in aromatic compounds highlighted in red), 1505 cm^−1^ (stretching vibration in aromatic ring highlighted in green), 1231 cm^−1^ (signal coming from ester bond groups and/or oxirane rings highlighted in blue), 1000–1050 cm^−1^ (doublet, signal of -CH2-OH grouping highlighted in gray) and 825 cm^−1^ (spring vibration of C-C bond in epoxy ring highlighted in brown).

The spectra of non-aged samples did not differ in the distribution of peaks, nor did they differ in intensity, which indicated a small effect of cellulose on the chemical properties of the composite. However, it should be noted that cellulose blurred the band in the range 1100–1000 cm^−1^, and signals without peaks in the range 600–400 cm^−1^ were visible and increased in intensity with increasing cellulose content in the sample.

When comparing the spectra of aged and non-aged samples, the greatest differences in peak intensities could be observed for samples subjected to thermo-oxidative aging. The intensity of the following peaks decreased significantly: 1505 cm^−1^ (possible opening of aromatic rings during free radical degradation), 1232 cm^−1^ and 825 cm^−1^.

The smallest changes in peak intensities were observed for the resin sample containing 10 phr cellulose, which may indicate an increased resistance of this composite to aging under the conditions studied.

The most visible signs of degradation occurring were in the sample containing 15 phr of cellulose after thermo-oxidative aging. There was a significant decrease in the intensity of peaks.

The decrease in the intensity of the above peaks indicated the occurrence of a degradation process with the opening of aromatic rings and oxirane rings remaining in the crosslinked resin.

At 20 phr cellulose content in the studied sample, after thermo-oxidative aging, an increase in spectral intensity was observed in peaks at 1507 cm^−1^, 1234 cm^−1^ and 826 cm^−1^. These values corresponded to the presence of aromatic and epoxy rings. The higher intensity of these peaks may indicate that the plate that was subjected to climatic aging may not have crosslinked to the same extent as the other samples. The presence of non-crosslinked resin in the test sample would explain the increased intensity of the peaks in question, even against a non-aged reference sample.

IR spectra showing changes occurring in the reference sample as well as in samples filled with 10 and 20 phr of cellulose are shown in Figure 7.

### 3.5. Overview and Discussion of TGA Results

Obtained mass loss temperatures and appropriate TGA curves are presented in Table 3 and Figure 8. TGA measurements were carried out on unaged samples of A.S. SET 1010 thermosetting epoxy resin with varying cellulose content to determine the thermal stability of said composites. First, mass loss was 2%, which corresponded to moisture content within the composites. Based on the obtained results and curves, the mass loss temperatures T_02_, T_10_, T_50_ and T_90_ corresponding to 2, 10, 50 and 90% of the mass loss, respectively, were determined. These temperatures are presented in the table below (Table 2.)

The obtained curves had two distinct steps of mass loss rates. The first step, which started around 2% of the mass loss rate, corresponded to the beginning of the composite degradation process. The second step in the curves was observed at around 90% of the mass loss across samples, and it corresponded with afterburning with residual ash. Analyzing the T_02_–T_90_ temperature values obtained, it could be observed that the addition of cellulose to the AS. SET. 1010 improved the thermal stability of the composite in the first stage of combustion, which was well illustrated by the T_02_ temperatures. The lowest temperature of 2% weight loss was observed for the reference sample (T_02_ = 206 °C), and the highest for samples containing 5 and 10 phr cellulose (T_02_ = 308 °C). At temperatures above 308 °C, samples containing cellulose did not show better thermal stability compared to the reference sample. The mass loss temperatures T_10_ and T_50_ were lower for the cellulose-containing samples than for the reference sample.

It is also worth noting that no correlation was observed between the increase in the amount of cellulose in the composite and the weight loss temperatures. This may be because the test samples were not homogeneous, and the cellulose content of the TGA-tested samples may not reflect the amount of cellulose contained in the volume of the test tiles.

### 3.6. Particle Dispersion Analysis

Dispersion analysis was performed for composite samples containing 5, 10, 15 and 20 phr of cellulose. The area tested was 12,538.78 µm^2^. In order to perform the analysis, the number of observed particles was counted from three randomly selected parts of the sample. The results of the conducted analysis are presented in the form of a bar chart in Figure 9. Figure 10 presents photographs of the examined areas.

From the collected results, it can be observed that the sample containing 15 phr of cellulose obtained a good degree of dispersion, of 50.33 ± 13.91 particles in the area of 12,538.78 µm^2^. The dispersion of cellulose in the resin at a content of 5 to 10 phr was at a comparable level. We observe weak interactions between the hydrophilic cellulose and the hydrophobic resin, directly translating into a deteriorated dispersion of the filler itself. Mere drying did not thermally modify the cellulose enough to obtain adequate resin reinforcement. Ultimately, part of the resin should be reduced and replaced in part by cellulose fibers in order to maintain the functional properties of the composite, and at the same time have a positive impact on the natural environment. The performed optical tests allowed us only to determine the general degree of dispersion and distribution of cellulose in the resin.

## 4. Conclusions

Summarizing the research results, it can be noted that the addition of bio-cellulose to the one-component epoxy resin A.S. SET 1010 did not cause any negative changes in the mechanical properties of the tested composites. This may indicate that cellulose in the case of A.S. SET 1010 resin can be used to, among other things, reduce the cost of composite production. The addition of cellulose can increase the resistance of the material to solar radiation, but a color change study showed that the material is more susceptible to aging at elevated temperatures with increasing cellulose content. We suspect that in the first stages of aging, the bio-filler acts as an absorber of UV radiation during the weathering process. On the other hand, cellulose had no significant effect on the thermal properties of the composite, as evidenced by the similar values of the glass transition temperature and practically the same combustion courses in the same temperature ranges, which were observed during TGA testing. To show whether the addition of cellulose has a negative or positive effect on other properties of the composite, it is necessary to conduct further testing in the field of flammability or mechanical strength by increasing the amount of cellulose added to A.S. SET 1010 resin, starting from an addition of at least 20 phr of cellulose (the amount of cellulose that caused an increase in the resistance of the material to aging processes).

## Figures and Tables

**Figure 1 materials-15-03256-f001:**
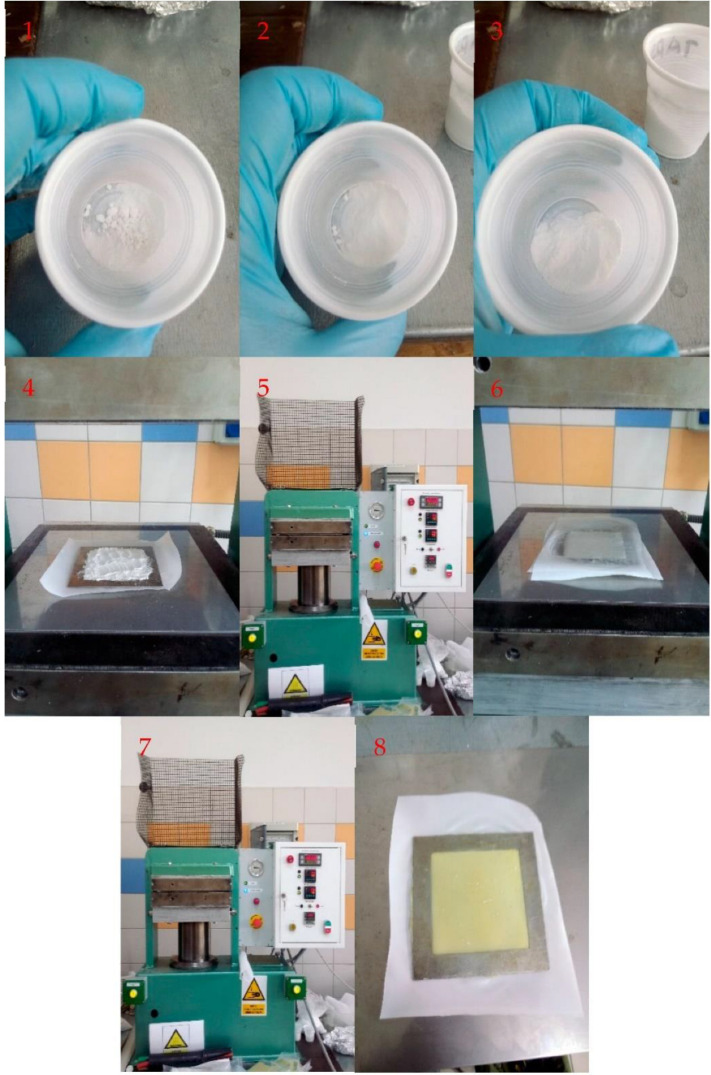
Preparation process for epoxy resin composites. (**1**–**3**) Mixing in batches in laboratory plastic cups. (**4**) Placing the material into the mold. (**5**) Plasticizing. (**6**) Heating the press plates to the crosslinking temperature. (**7**) Curing. (**8**) Removing the finished plate from the mold.

**Figure 2 materials-15-03256-f002:**
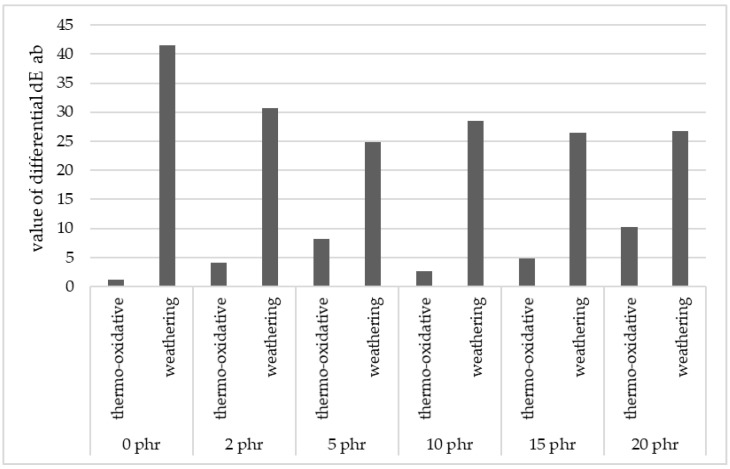
Color change of samples after simulated aging processes (comparison of dE ab values) against reference sample.

**Figure 3 materials-15-03256-f003:**
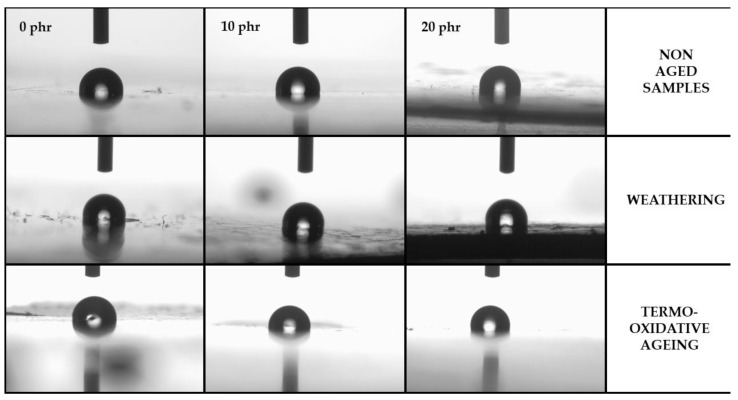
Comparison of distilled water droplet shapes on the surface of composites with increasing amounts of cellulose. In all of the above cases, the hydrophobic properties are clearly visible.

**Figure 4 materials-15-03256-f004:**
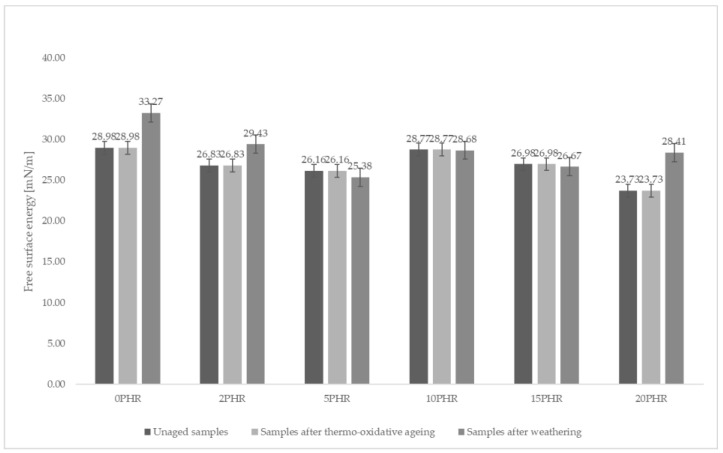
Surface energies of the composites studied. As can be seen, the highest values of surface energy were, in most cases, characterized by samples subjected to climatic aging (the highest for sample 0 phr = 33.27 mN/m). In all cases, the surface energies of samples subjected to thermo-oxidative aging did not differ from that of the non-aged specimen.

**Figure 5 materials-15-03256-f005:**
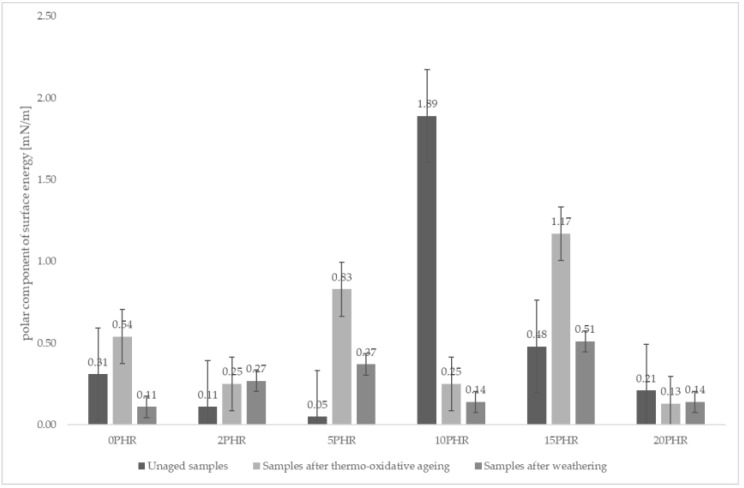
Comparison of the polar components of the surface energies of the samples studied. The sample containing 10 phr cellulose had the highest polar component of surface energy, 1.89 mN/m.

**Figure 6 materials-15-03256-f006:**
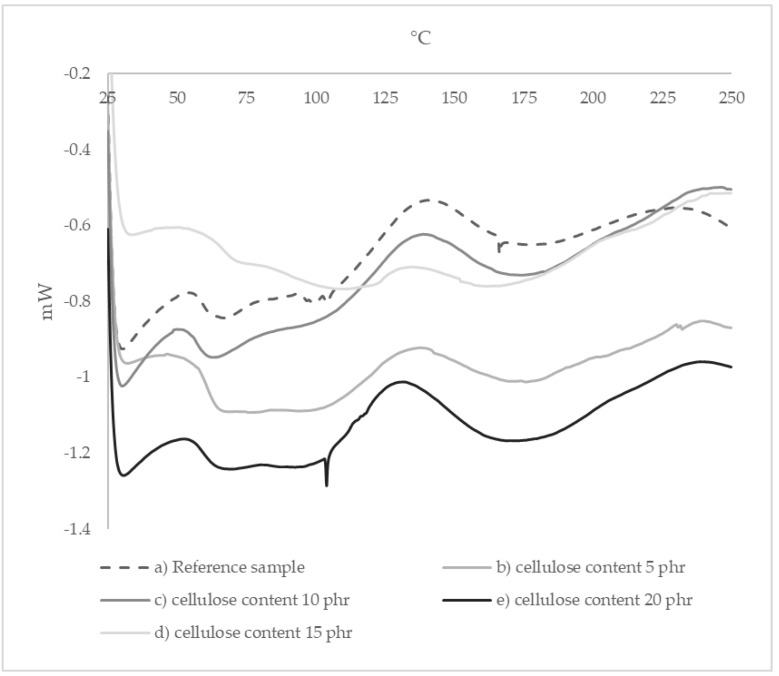
DSC graphs of composite samples with addition of 5, 10 and 20 phr of cellulose in comparison to pure epoxy resin (0 phr of cellulose).

**Figure 7 materials-15-03256-f007:**
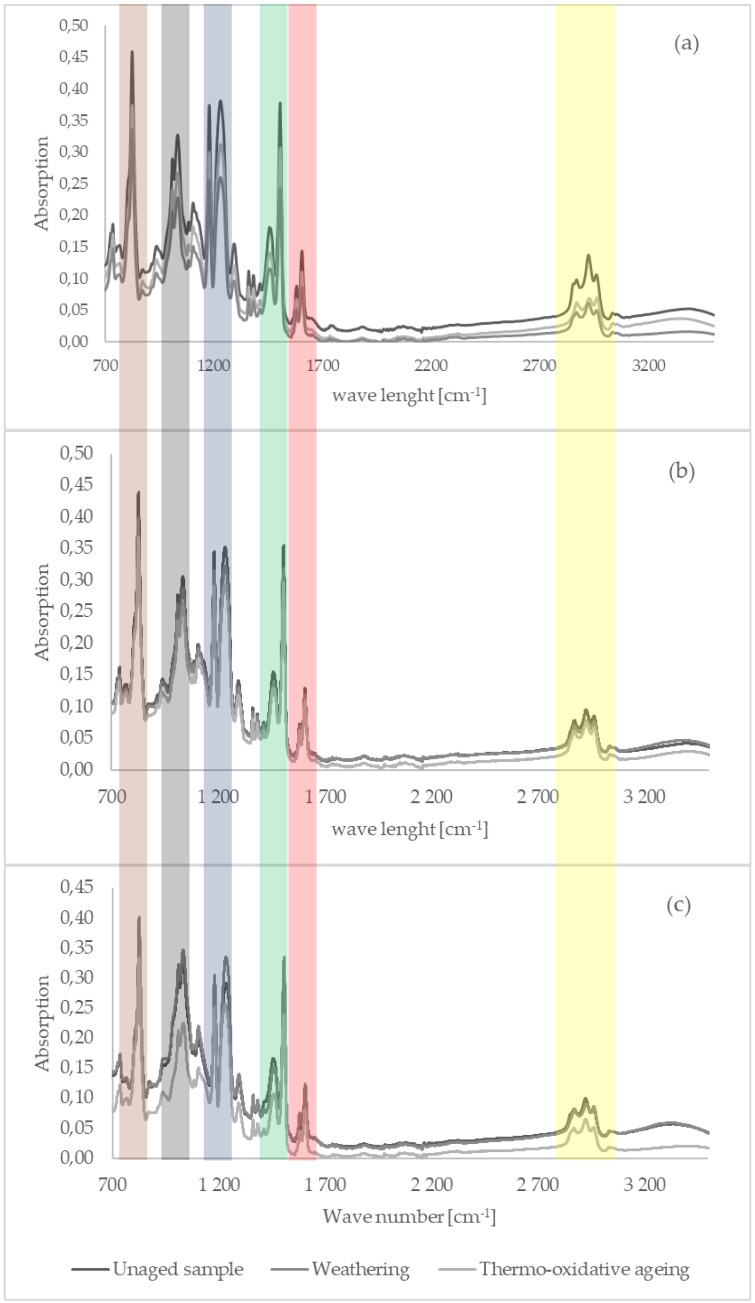
Comparison of the IR spectra of samples containing (**a**) 0 phr cellulose, (**b**) 10 phr cellulose and (**c**) 20 phr cellulose before and after simulated aging processes.

**Figure 8 materials-15-03256-f008:**
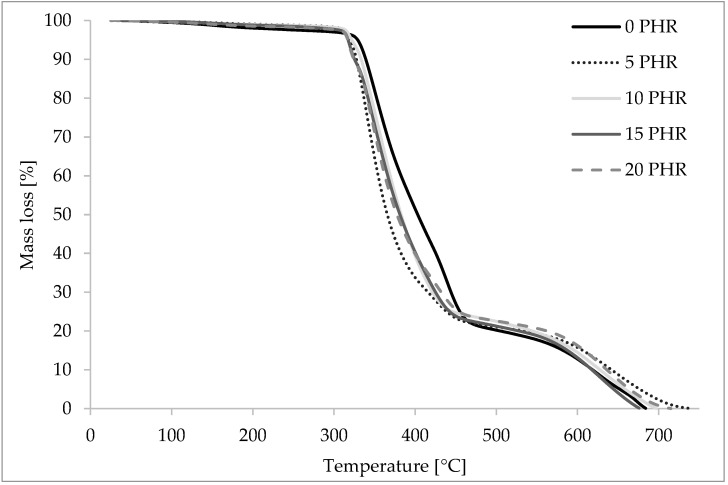
Thermogravimetric (TG) curves of unaged composite samples containing 5, 10, 15 and 20 phr cellulose. The reference point is the curve for the sample containing no cellulose and is marked with a continuous black line. As can be seen, the curves are close to the reference curve.

**Figure 9 materials-15-03256-f009:**
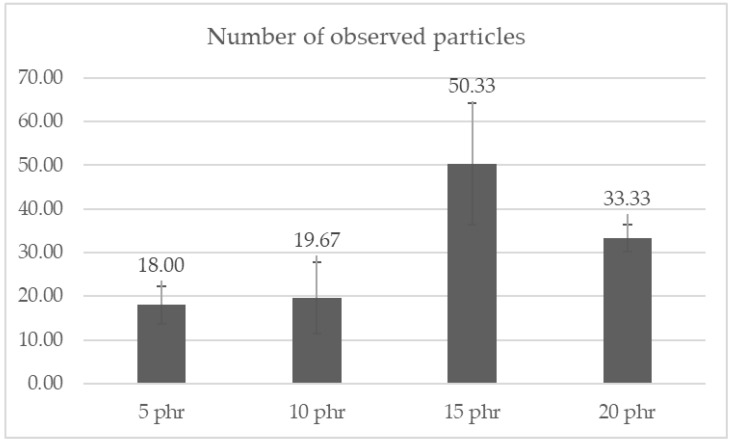
Graph showing the number of observed cellulose particles in the composites tested (with 5, 10, 15 and 20 phr cellulose content).

**Figure 10 materials-15-03256-f010:**
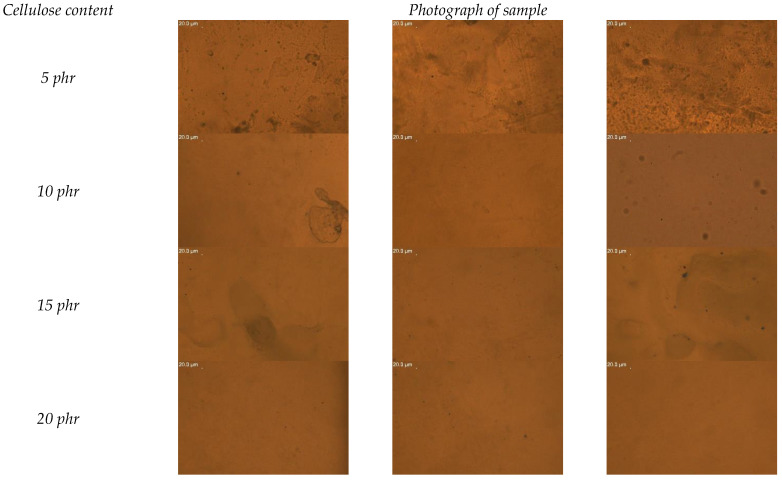
Microscopic images comparing the degree of cellulose dispersion in the samples of composites with 5, 10, 15 and 20 phr cellulose content.

**Table 1 materials-15-03256-t001:** Color change after simulated aging process in CIE-Lab coordinate system.

Amount of Cellulose	Type of Aging Process	L	a	b	dL	da	db	dE ab
**0 phr**	thermo-oxidative	88.83	−3.84	14.63	−0.65	0.67	−0.69	1.16
	weathering	66.05	11.38	45.66	23.43	15.89	30.34	41.49
**2 phr**	thermo-oxidative	77.73	−269.00	18.86	−0.12	0.97	−4.03	4.15
	weathering	55.12	11.38	36.03	27.73	15.04	14.14	30.70
**5 phr**	thermo-oxidative	74.98	−2.95	19.29	3.48	1.65	−7.19	8.15
	weathering	53.69	11.15	33.52	17.81	15.75	7.05	24.80
**10 phr**	thermo-oxidative	72.02	−4.14	23.22	1.40	0.23	−2.26	2.66
	weathering	47.44	12.22	24.60	23.18	16.58	−0.87	28.51
**15 phr**	thermo-oxidative	70.51	−3.48	20.36	−0.02	0.45	−4.80	4.82
	weathering	48.87	10.86	22.14	21.67	14.79	−3.01	26.40
**20 phr**	thermo-oxidative	74.44	−3.54	23.60	4.20	−0.05	−9.28	10.18
	weathering	52.53	9.04	17.19	17.72	12.54	15.69	26.78

**Table 2 materials-15-03256-t002:** Thermal changes in A.S. SET 1010 resin samples with 0, 5, 10, 15 and 20 phr cellulose obtained using DSC analysis.

CELLULOSE CONTENT [phr]	TEMPERATURE [°C]				
	ONSET 1	GLASS TRANSITION	ONSET 2	PEAK	ENDSET
**0**	56.34	60.42	104.45	136.31	165.88
**5**	58.04	61.71	109.77	136.96	154.24
**10**	54.68	58.69	108.35	137.14	165.96
**15**	63.77	68.32	121.36	134.80	153.81
**20**	56.34	61.61	104.71	130.12	160.90

**Table 3 materials-15-03256-t003:** Mass loss temperatures (TG) of unaged epoxy resin composites with different cellulose content.

[°C]	REF	5 phr	10 phr	15 phr	20 phr
**T_02_**	206	308	308	286	277
**T_10_**	339	327	333	325	324
**T_50_**	403	365	382	381	377
**T_90_**	618	642	627	618	636

## Data Availability

No new data were created or analyzed in this study. Data sharing is not applicable to this article.

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
