# Peer review of "Accelerated Aging of Epoxy Biocomposites Filled with Cellulose"

_materials, 2022, doi:10.3390/ma15093256_

Round 1

Reviewer 1 Report

In the paper under consideration (named as “Accelerated ageing of epoxy biocomposites filled with cellulose”), the authors studied influence of biomass cellulose on the ageing process of epoxy resin. This article is not innovative enough but has its scientific topic to a certain extent. Some sentences are confusing and the organization lacks logic. Therefore, I recommend making a decision after the authors have made major revisions to this manuscript.

  1. “positive impact on the natural environment.” How can these impacts be assessed?
  2. Reference number shouldn’t be placed at the beginning of a sentence
  3. Why cellulose powder can be used to prepare reinforced epoxy resin composites? The mechanical properties of epoxy resin are stronger than cellulose and the compatibility between them is very poor. The author said some surface modification methods of cellulose, why the authors didn’t choose modified cellulose powder?
  4. “This is an innovative solution for railway industry”. There are many literatures about epoxy resin/cellulose composites. The novelty stated by authors is not innovative enough.
  5. The logicality of “Introduction” section is very poor, especially from the second paragraph to the fifth paragraph. Please ask some professionals for help and reorganize it.
  6. “The main objective of this research is to increase the ageing susceptibility of applied resin composites”. How did the authors prove that the increased ageing susceptibility didn’t “losing the initially good functional properties”?
  7. In Introduction section, “Innovatively, modified cellulose fibers were used”. In 2.1 section, “filled with 0, 2, 5, 10, 15 and 20 phr of dried, unmodified cellulose powder”. Are fibers or powders used in this work, modified or unmodified?
  8. “there is a decreasing trend of colour change with increasing cellulose content can be observed.” In weather type of ageing process, why did the sample containing 5phr cellulose show the lowest differential dE*ab?
  9. Grammar check. “Discussion of obtained DSC curvrs” “Table 2. Observed thermal changes during DSC analysis in A.S. SET 1010 resin samples containing 0, 5, 10, 15 and 20 phr cellulos.e” “presented in Table 2. and Figure 4.” “T02 = 206°C”
  10. “this sample was much better crosslinked at the forming stage than the other samples because the exothermic peak”. Why and how epoxy resin with 15 phr cellulose had the best crosslinked state? Since all samples were not completely crosslinked, can the author provide the degree of cross-linking of each sample? If not, does it mean that the authors evaluated and compared the performance of the samples under different conditions?
  11. The format of Section 3.4 in PDF version is disordered. It is very difficult to read. Please reorganize it.
  12. Please provide TGA results of aged composites, therefore, the influence of ageing processes on the thermal stability of epoxy resin with different amount celluloses can be evaluated.
  13. “This may be because the test samples were not homogeneous, and the cellulose content of the TGA-tested samples may not reflect the amount of cellulose contained in the volume of the test tiles.” If so, the results and conclusions obtained from FTIR and DSC tests also can’t properly reflect the real properties of samples. Please characterize the dispersion state of celluloses in epoxy resin matrix through SEM or other techniques.

Author Response

Institute of Polymer and Dye Technology

Technical University of Lodz

90-924 Lodz, ul Stefanowskiego 12/16, Poland

Tel.: +48 42 631 32 23, Fax: +48 42 636 25 43

March 21st, 2022

Materials (MDPI)

Dear Editors,

We are resubmitting our revised paper entitled Accelerated ageing of epoxy biocomposites filled with cellulose by Anna Masek, Radosław Busiak, Aleksandra Węgier and Adam Rylski with a request to reconsider it for publication in Materials (MDPI)

We have carefully considered the Reviewers comments. The manuscript was revised according to these comments. The list of responses to the reviewers comments and corrections made in the manuscript is attached.

The manuscript has not been previously published, is not currently submitted for review to any other journal and will not be submitted elsewhere before a decision is made by this journal.

For correspondence, please use the following information:

corresponding author: Anna Masek

Institute of Polymer and Dye Technology

Technical University of Lodz

90-924 Lodz, ul Stefanowskiego 12/16, Poland

Tel.: +48 42 631 32 93

Fax: +48 42 636 25 43

e-mail: anna.masek@p.lodz.pl

Yours sincerely,

Ph. D., D.Sc. Anna Masek

Reviewer 1:

  1. “positive impact on the natural environment.” How can these impacts be assessed?

Our research work is still ongoing we envisage tests such as biodegradability and compostability. In general, the addition of cellulose should increase the hydrophilicity of the material and hence the susceptibility to degradation by microorganisms. In general, the lifetime of the natural fiber reinforced resin should be shorte,r than that of the unmodified material.

  1. Reference number shouldn’t be placed at the beginning of a sentence

Thank you for drawing attention to this error. It has been considered and corrected.

  1. Why cellulose powder can be used to prepare reinforced epoxy resin composites? The mechanical properties of epoxy resin are stronger than cellulose and the compatibility between them is very poor. The author said some surface modification methods of cellulose, why the authors didn’t choose modified cellulose powder?

In our research we used very short cellulose fibers, literally describing cellulose powder. Fiber hydrophobization is based on a simple mechanochemical process. Codification therefore to make it easy to transfer the process to an industrial scale. The research is carried out in connection with the implementation project in cooperation with a company that manufactures equipment for train rolling stock. Ultimately, the epoxy resin is reinforced with glass fibers and the addition of cellulose fibers is treated as a biofiller aimed at shortening the life of the products and increasing their susceptibility to degradation.

  1. “This is an innovative solution for railway industry”. There are many literatures about epoxy resin/cellulose composites. The novelty stated by authors is not innovative enough.

The scientific novelty is sought in the very mechanochemical modification / hydrophobization of cellulose fibers. An innovative approach to testing modified resin samples and the use of this type of materials for the railway industry. First of all, these tests are a novelty in the implementation of the in railway industry, as evidenced by the project implemented in cooperation with a company dealing in the production of rolling stock for trains.

  1. The logicality of “Introduction” section is very poor, especially from the second paragraph to the fifth paragraph. Please ask some professionals for help and reorganize it.

Thank you for this comment. This section of the publication has been modified accordingly.  

  1. “The main objective of this research is to increase the ageing susceptibility of applied resin composites”. How did the authors prove that the increased ageing susceptibility didn’t “losing the initially good functional properties”?

This is our overall goal in research. However, we would like to point out that in both the abstract and the conclusions of our results we stated that cellulose, added to researched epoxy resin, in amounts up to 20 phr did not inflicted any significant changes within studied parameters.

However, this comment drew our attention to the fact that this statement in the introduction could be misleading to the reader.

  1. In Introduction section, “Innovatively, modified cellulose fibers were used”. In 2.1 section, “filled with 0, 2, 5, 10, 15 and 20 phr of dried, unmodified cellulose powder”. Are fibers or powders used in this work, modified or unmodified?

Thank you for this comment. We have improved the vocabulary in the publication so that it is more consistent and specific.

  1. There is a decreasing trend of colour change with increasing cellulose content can be observed.” In weather type of ageing process, why did the sample containing 5phr cellulose show the lowest differential dE*ab?

Although the lowest value was obtained for the sample containing 5phr cellulose, a general downward trend can also be observed in the graph of colour variations in Figure 2.

  1. Grammar check. “Discussion of obtained DSC curvrs” “Table 2. Observed thermal changes during DSC analysis in A.S. SET 1010 resin samples containing 0, 5, 10, 15 and 20 phr cellulos.e” “presented in Table 2. and Figure 4.” “T02 = 206°C”

We took your suggestion into consideration, and we rephrased titles of Table 2 and Figure 4 to be grammatically correct.

  1. “this sample was much better crosslinked at the forming stage than the other samples because the exothermic peak”. Why and how epoxy resin with 15 phr cellulose had the best crosslinked state? Since all samples were not completely crosslinked, can the author provide the degree of cross-linking of each sample? If not, does it mean that the authors evaluated and compared the performance of the samples under different conditions?

At the time of reaserch we did not tested degree of cross-linking of the samples. Regardless, all samples were prepared and treated in exactly same conditions. With that said, the crosslinking peak that could be atributed to crosslinking of the resin in tested sample (with 15 phr content of cellulose) may be coused by factor outside our control.

  1. The format of Section 3.4 in PDF version is disordered. It is very difficult to read. Please reorganize it.

Thank you for your comment. the indicated section of the PDF version of the document will be reorganised.

  1. Please provide TGA results of aged composites, therefore, the influence of ageing processes on the thermal stability of epoxy resin with different amount celluloses can be evaluated.

At the time of the study, we considered it unnecessary to test aged samples. Unfortunately, we are unable to test the aged samples again because we no longer have these samples.

  1. “This may be because the test samples were not homogeneous, and the cellulose content of the TGA-tested samples may not reflect the amount of cellulose contained in the volume of the test tiles.” If so, the results and conclusions obtained from FTIR and DSC tests also can’t properly reflect the real properties of samples. Please characterize the dispersion state of celluloses in epoxy resin matrix through SEM or other techniques.

Unfortunately, we do not have the possibility and necessary equipment to test the degree of dispersion at this time. However, we will take this into account in future studies to provide the most accurate results.

Reviewer 2 Report

Note: for all questions, the intent is to answer them in the manuscript and not answer me.  Additionally, the grammar is not well prepared in many places.

  1. In the abstract, what does the acronym phr stand for?
  2. Abstract line 20 – what properties were not significantly affected? Conversely, I thought the addition of cellulose was to improve properties?  There does not seem to be value in not changing.
  3. It is said in line 40 that “Metal parts, despite their excellent mechanical properties, are characterized by high weight and price.” What is lacking in this discussion is the life cycle benefits of cellulose materials (commonly solid wood or nanocellulose too) over steel. 
  4. Line 46 – is epoxy low price? I think this is relative and sometimes is considered low and sometimes is considered expensive depending on the alternative.
  5. Line 52 – dispersion of cellulose is also hard during mixing due to viscosity increases.
  6. Lines 56-60 promotes this research as being important for railcar seats; however, the research is around weathering and color change. I am not debating that this is not important, but I am curious if this is the most important metrics to measure for comparison?
  7. Table 1 – what does phr mean?
  8. Line 62 – you say cellulose is fully degradable. Is it still degradable after cross linking with epoxy? 
  9. The degradation of cellulose to other events such as rot fungi should be mentioned. Yang, Yan, et al. "Fourier-Transform Infrared Spectroscopy Analysis of the Changes in Chemical Composition of Wooden Components: Part II—The Ancient Building of Danxia Temple." Forest Products Journal3 (2021): 283-289. In that study, the crystallinity also changed with degradation.  I do not see any mention of crystallinity.  But if you look at Yang, Tan et al. listed above, they used the ratio of the crystalline to amorphous region in the FTIR spectra to identify this degradation in the crystalline structure.
  10. 3 – why show the actual contact angle data instead of just saying everything was greater than 90? This seems to devalue the work.
  11. Table 2 – Can you really measure temperature to the 0.01 precision? As a reader, I find it distracting when someone says 56.34 degrees.
  12. 6 how did the software connect the dots to make a smooth line? Data fitting method?
  13. The references do not include any forest product journal type papers. With the discussion of cellulose, I would have expected some of the more “local” discipline journals as opposed to just high impact factor journals.

Author Response

Institute of Polymer and Dye Technology

Technical University of Lodz

90-924 Lodz, ul Stefanowskiego 12/16, Poland

Tel.: +48 42 631 32 23, Fax: +48 42 636 25 43

March 21st, 2022

Materials (MDPI)

Dear Editors,

We are resubmitting our revised paper entitled Accelerated ageing of epoxy biocomposites filled with cellulose by Anna Masek, Radosław Busiak, Aleksandra Węgier and Adam Rylski with a request to reconsider it for publication in Materials (MDPI)

We have carefully considered the Reviewers comments. The manuscript was revised according to these comments. The list of responses to the reviewers comments and corrections made in the manuscript is attached.

The manuscript has not been previously published, is not currently submitted for review to any other journal and will not be submitted elsewhere before a decision is made by this journal.

For correspondence, please use the following information:

corresponding author: Anna Masek

Institute of Polymer and Dye Technology

Technical University of Lodz

90-924 Lodz, ul Stefanowskiego 12/16, Poland

Tel.: +48 42 631 32 93

Fax: +48 42 636 25 43

e-mail: anna.masek@p.lodz.pl

Yours sincerely,

Ph. D., D.Sc. Anna Masek

Reviewer 2:

  1. and 7. In the abstract, what does the acronym phr stand for? / 7. Table 1 – what does phr mean?

PHR stands for Per Hundred of Resin. Also, acronym was replaced in abstract.

  1. Abstract line 20 – what properties were not significantly affected? Conversely, I thought the addition of cellulose was to improve properties? There does not seem to be value in not changing.

Our research in, in this paper, focuses on surface and physical changes to the composite. We were anticipated changes in surface properties but results within tested parameters proven to be insignificant.

  1. It is said in line 40 that “Metal parts, despite their excellent mechanical properties, are characterized by high weight and price.” What is lacking in this discussion is the life cycle benefits of cellulose materials (commonly solid wood or nanocellulose too) over steel.

Thank you for this comment. Our main point of focus were composite materials with biobased fillers – in case of this paper cellulose. We did not consider discussing materials than steel, because that would not fit context set in introduction. However, we will take that comment into consideration and discuss researched  material in broader context in our subsequent work.

  1. Line 46 – is epoxy low price? I think this is relative and sometimes is considered low and sometimes is considered expensive depending on the alternative.

In the context of railway industry, that was mentioned in the introduction section, epoxy resin is cheaper alternative to steal and aluminum.

  1. Line 52 – dispersion of cellulose is also hard during mixing due to viscosity increases.

Thank you for this valuable comment.

  1. Lines 56-60 promotes this research as being important for railcar seats; however, the research is around weathering and color change. I am not debating that this is not important, but I am curious if this is the most important metrics to measure for comparison?

Extensive mechanical testing of this type of composites are being conducted and will be published in another manuscript. Due to amount of data collected we decided to split them into two separate manuscripts. In this particular case we focused on surface changes and thermal stability of one component epoxy resin A.S. SET 1010 filled with cellulose.

  1. Line 62 – you say cellulose is fully degradable. Is it still degradable after cross linking with epoxy?

Based on the results obtained, it can be concluded that the composite of epoxy resin and cellulose may have difficulty with biodegradation in the environment. However to answer this question it is necessary to carry out accelerated ageing tests under composting conditions or in the presence of microorganisms.

  1. Why show the actual contact angle data instead of just saying everything was greater than 90? This seems to devalue the work.

We believe that it is not necessary to present all the obtained values of wetting angles. From our point of view, the most important results are the obtained values of surface energies shown in Figure 4 and Figure 5.

  1. Table 2 – Can you really measure temperature to the 0.01 precision? As a reader, I find it distracting when someone says 56.34 degrees.

MetlerToledo DSC/TGA1 aparatus used for TGA and DSC testing can measure temperature with accuracy up to 0.01°C. However, thank you for your attention to how our results present themselves to the reader. 

  1. How did the software connect the dots to make a smooth line? Data fitting method?

Presented graphs are point graphs with smoothen lines prepared in MS Excel. Number of obtained data points allowed us to prepare smooth and easy to read graphs. But we are willing change type of graphs to only point graphs if necessary to increase accuracy of presented data.

  1. The references do not include any forest product journal type papers. With the discussion of cellulose, I would have expected some of the more “local” discipline journals as opposed to just high impact factor journals.

We had no criteria for the selection of sources when preparing the publication. However, we would like to point out that several technical journals were listed among our sources. We thank you for this valuable remark and will try to use local sources more often in our future work.

Round 2

Reviewer 1 Report

The author made some revisions and improvements to this article but did not give reasonable answers to some of the questions raised by the reviewers. In particular, no substantive solutions have been made to some key issues such as TGA testing and characterization of cellulose dispersion. Authors are advised to revise the paper again based on the comments.

  1. “positive impact on the natural environment.” How can these impacts be assessed?

Our research work is still ongoing we envisage tests such as biodegradability and compostability. In general, the addition of cellulose should increase the hydrophilicity of the material and hence the susceptibility to degradation by microorganisms. In general, the lifetime of the natural fiber reinforced resin should be shorte,r than that of the unmodified material.

  • According to the Figure 3 and the statement in the manuscript (“On the basis of obtained results, it can be observed that all tested samples pose hydrophobic properties (the values of contact angles for water droplets are higher than 90°). Moreover, the content of dispersed cellulose in the whole volume of the resin has no visible effect on the surface properties of the material.”), the hydrophilicity of samples changed little. How will these small changes have a major impact on the subsequent biodegradation process?

  1. Why cellulose powder can be used to prepare reinforced epoxy resin composites? The mechanical properties of epoxy resin are stronger than cellulose and the compatibility between them is very poor. The author said some surface modification methods of cellulose, why the authors didn’t choose modified cellulose powder?

In our research we used very short cellulose fibers, literally describing cellulose powder. Fiber hydrophobization is based on a simple mechanochemical process. Codification therefore to make it easy to transfer the process to an industrial scale. The research is carried out in connection with the implementation project in cooperation with a company that manufactures equipment for train rolling stock. Ultimately, the epoxy resin is reinforced with glass fibers and the addition of cellulose fibers is treated as a biofiller aimed at shortening the life of the products and increasing their susceptibility to degradation.

  • What is “very short”? Please provide the size of powder fibres.
  • What is “simple mechanochemical process”? Please explain it.
  • Codification?
  • As the authors mentioned in their answer to the 1st question, the addition of cellulose will “increase the hydrophilicity”, thereby increase “the susceptibility to degradation by microorganisms”. So, why were hydrophobized fibres used in this work?

  1. “This is an innovative solution for railway industry”. There are many literatures about epoxy resin/cellulose composites. The novelty stated by authors is not innovative enough.

The scientific novelty is sought in the very mechanochemical modification / hydrophobization of cellulose fibers. An innovative approach to testing modified resin samples and the use of this type of materials for the railway industry. First of all, these tests are a novelty in the implementation of the in railway industry, as evidenced by the project implemented in cooperation with a company dealing in the production of rolling stock for trains.

  • There is no “very mechanochemical modification / hydrophobization of cellulose fibers” in this work. The authors did not provide any results about the mechanochemical modification or hydrophobization results of cellulose fibres.
  • Which is the “innovative approach to testing modified resin samples and the use of this type of materials for the railway industry”? All test methods used in this work are very common. The field of application (railway) may be different from other EP/cellulose composites, but it is not new. In addition, there are no results directly related to real railway conditions in the manuscript.
  1. There is a decreasing trend of colour change with increasing cellulose content can be observed.” In weather type of ageing process, why did the sample containing 5phr cellulose show the lowest differential dE*ab?

Although the lowest value was obtained for the sample containing 5phr cellulose, a general downward trend can also be observed in the graph of colour variations in Figure 2.

  • Why did the sample containing 5phr cellulose show the lowest differential dE*ab?
  1. “this sample was much better crosslinked at the forming stage than the other samples because the exothermic peak”. Why and how epoxy resin with 15 phr cellulose had the best crosslinked state? Since all samples were not completely crosslinked, can the author provide the degree of cross-linking of each sample? If not, does it mean that the authors evaluated and compared the performance of the samples under different conditions?

At the time of reaserch we did not tested degree of cross-linking of the samples. Regardless, all samples were prepared and treated in exactly same conditions. With that said, the crosslinking peak that could be atributed to crosslinking of the resin in tested sample (with 15 phr content of cellulose) may be coused by factor outside our control.

  • reaserch”, “atribute”, “coused”? While these responses will not appear in your publication, the authors should take it seriously!
  • Please answer the question “Why and how epoxy resin with 15 phr cellulose had the best crosslinked state?”
  • What is the “factor outside our control”?
  1. Please provide TGA results of aged composites, therefore, the influence of ageing processes on the thermal stability of epoxy resin with different amount celluloses can be evaluated.

At the time of the study, we considered it unnecessary to test aged samples. Unfortunately, we are unable to test the aged samples again because we no longer have these samples.

  • This manuscript investigated “Accelerated ageing of epoxy biocomposites filled with cellulose”, in my opinion, it is necessary to do this test to investigate the influence of ageing processes on the thermal stability of epoxy resin.
  • “no longer have these samples” is not a reason in this situation. What you did is a repeatable scientific work, not a one-shot deal.
  1. “This may be because the test samples were not homogeneous, and the cellulose content of the TGA-tested samples may not reflect the amount of cellulose contained in the volume of the test tiles.” If so, the results and conclusions obtained from FTIR and DSC tests also can’t properly reflect the real properties of samples. Please characterize the dispersion state of celluloses in epoxy resin matrix through SEM or other techniques.

Unfortunately, we do not have the possibility and necessary equipment to test the degree of dispersion at this time. However, we will take this into account in future studies to provide the most accurate results.

  • The dispersion of cellulose or the uniformity of EP/cellulose composite plays an important role in the analysis of TGA, DSC, and FTIR results. If the composites fabricated in this work are not homogeneous, the credibility of the analysis results (TGA, DSC, and FTIR) of this article will be greatly reduced. If the authors can not provide any results about it, frankly speaking, this work may not meet scientific requirements about this journal.

Author Response

Institute of Polymer and Dye Technology

Technical University of Lodz

90-924 Lodz, ul Stefanowskiego 12/16, Poland

Tel.: +48 42 631 32 23, Fax: +48 42 636 25 43

April 8th, 2022

Materials (MDPI)

Dear Editors,

We are resubmitting our revised paper entitled Accelerated ageing of epoxy biocomposites filled with cellulose by  Radosław Busiak, Anna Masek, Aleksandra Węgier and Adam Rylski with a request to reconsider it for publication in Materials (MDPI)

We have carefully considered the Reviewers comments. The manuscript was revised according to these comments. The list of responses to the reviewers comments and corrections made in the manuscript is attached.

The manuscript has not been previously published, is not currently submitted for review to any other journal and will not be submitted elsewhere before a decision is made by this journal.

For correspondence, please use the following information:

corresponding author: Anna Masek

Institute of Polymer and Dye Technology

Technical University of Lodz

90-924 Lodz, ul Stefanowskiego 12/16, Poland

Tel.: +48 42 631 32 93

Fax: +48 42 636 25 43

e-mail: anna.masek@p.lodz.pl

Yours sincerely,

Ph. D., D.Sc. Anna Masek

We would like to address your questions and comments on the manuscript and the response.

The author made some revisions and improvements to this article but did not give reasonable answers to some of the questions raised by the reviewers. In particular, no substantive solutions have been made to some key issues such as TGA testing and characterization of cellulose dispersion. Authors are advised to revise the paper again based on the comments.

  1. “positive impact on the natural environment.” How can these impacts be assessed?

Our research work is still ongoing we envisage tests such as biodegradability and compostability. In general, the addition of cellulose should increase the hydrophilicity of the material and hence the susceptibility to degradation by microorganisms. In general, the lifetime of the natural fiber reinforced resin should be shorte,r than that of the unmodified material.  

According to the Figure 3 and the statement in the manuscript (“On the basis of obtained results, it can be observed that all tested samples pose hydrophobic properties (the values of contact angles for water droplets are higher than 90°). Moreover, the content of dispersed cellulose in the whole volume of the resin has no visible effect on the surface properties of the material.”), the hydrophilicity of samples changed little. How will these small changes have a major impact on the subsequent biodegradation process?

Answer: Thank You for comment, we agree with the review. Yes, the impact on the surfaces is not great due to the high level of hydrophobicity of the resin itself. However, the use of 15 to 20 phr of cellulose, typically a hydrophilic and natural filler, will certainly not remain indifferent to the degradation process of the entire sample. Despite the fact that there are no significant changes in the goniometric tests. The addition of, for example, 20 phr of cellulose is aimed at reducing the consumption of synthetic epoxy resin and replacing it in a percentage in the composite. Therefore, any reduction in the content of non-degradable polymer should have a positive effect on the environment.

  1. Why cellulose powder can be used to prepare reinforced epoxy resin composites? The mechanical properties of epoxy resin are stronger than cellulose and the compatibility between them is very poor. The author said some surface modification methods of cellulose, why the authors didn’t choose modified cellulose powder?

The aim of our work is to investigate the properties of an epoxy resin composite with natural, unmodified cellulose with very short fibres. The emphasis of our work is on the use of modern simulated ageing techniques to prepare a composite with as little environmental impact as possible after service lifetime.

  1. “This is an innovative solution for railway industry”. There are many literatures about epoxy resin/cellulose composites. The novelty stated by authors is not innovative enough.

The scientific novelty is sought in the very mechanochemical modification / hydrophobization of cellulose fibers. An innovative approach to testing modified resin samples and the use of this type of materials for the railway industry. First of all, these tests are a novelty in the implementation of the in railway industry, as evidenced by the project implemented in cooperation with a company dealing in the production of rolling stock for trains.

  • There is no “very mechanochemical modification / hydrophobization of cellulose fibers” in this work. The authors did not provide any results about the mechanochemical modification or hydrophobization results of cellulose fibres.
  • Which is the “innovative approach to testing modified resin samples and the use of this type of materials for the railway industry”? All test methods used in this work are very common. The field of application (railway) may be different from other EP/cellulose composites, but it is not new. In addition, there are no results directly related to real railway conditions in the manuscript.

Answer: We apologize for the mistake, the description related to the next tests that will already include cellulose modification. In this publication, we only show the effect of dried cellulose fibers of a certain length. It is very difficult to answer on the novelty itself, as there is currently a lot of research being done on application-related cellulose fibers. Certainly, the very novelty is the introduction of microcrystalline, ultra-fine and highly pure cellulose powder into the epoxy resin and the tests of accelerated aging.

  1. Please provide the size of powder fibres.

Particle size of used cellulose was in range 6-12µm in accordance with technical datasheet provided by manufacturer.  ARBOCEL® UFC 100 by JRS Rettenmaier is a microcrystalline, ultra-fine and highly pure cellulose powder. It acts as thickener. The small particle size and purity makes it a high performance filler for diverse coating applications. It is an environment friendly grade and is gained from replenish-able raw materials. It shows heat resistance and thixotrophy. ARBOCEL® UFC 100 is used as retention aid in paper coatings and volume agent in textile surfaces application.

  1. What is “simple mechanochemical process”?

Our aim was to make a biocomposite (as defined as a plastic composite with a natural filler) and to investigate its end-of-life properties using simulated ageing techniques. Nevertheless, a reviewer's comment drew our attention to the fact that the phrase "simple mechanochemical process" appeared in the abstract. The abstract has been appropriately modified to avoid misleading the potential reader.

  1. As the authors mentioned in their answer to the 1st question, the addition of cellulose will “increase the hydrophilicity”, thereby increase “the susceptibility to degradation by microorganisms”. So, why were hydrophobized fibres used in this work?

Cellulose fibres are hydrophobized to improve the compatibility of the filler with the polymer matrix. However, we would like to point out that even the hydrophobized cellulose surface is much more hydrophilic than the polymer matrix. 

  1. “This is an innovative solution for railway industry”. There are many literatures about epoxy resin/cellulose composites. The novelty stated by authors is not innovative enough.

An aspect of novelty of our work is the use of cellulose powder with very short fibres (6-12µm), not previously used in this type of biocomposites, and the performance of extensive ageing studies.

  1. There is no “mechanochemical modification / hydrophobization of cellulose fibers” in this work. The authors did not provide any results about the mechanochemical modification or hydrophobization results of cellulose fibres.

Like we presented in point 3 of this answer, we did not modify cellulose powder for this experiment. Therefore, we do not have results and descriptions of cellulose modifications in the description of the experiment.

  1. Which is the “innovative approach to testing modified resin samples and the use of this type of materials for the railway industry”? All test methods used in this work are very common. The field of application (railway) may be different from other EP/cellulose composites, but it is not new. In addition, there are no results directly related to real railway conditions in the manuscript.

As we stated in our answer in section 5, we believe that the novelty is not in epoxy resin biocomposite with cellulose, but in the use of natural cellulose with very short fibers and the performance of extensive ageing tests to determine the properties of the material after its lifetime in order to assess the environmental impact of such a material after its exploitation.

  1. There is a decreasing trend of colour change with increasing cellulose content can be observed.” In weather type of ageing process, why did the sample containing 5phr cellulose show the lowest differential dE*ab?

We wish to clarify the issue of this colour change result for the sample containing 5phr cellulose, as well as for the whole measurement and the nature of its result. Analysing the results obtained, it can be seen that the thermo-oxidative ageing of the samples resulted in approximately 3 times more colour change than the weathering. Furthermore, a trend of decreasing intensity of colour change due to ageing with increasing cellulose content can be observed. In the case of the colour change after weathering, the differences between the results obtained are small and within the measuring error of the apparatus used (approximately 20%). At this point we would like to emphasise that the results presented are of a qualitative rather than quantitative nature.

  1. Why did the sample containing 5phr cellulose show the lowest differential dE*ab?

The issue in question is discussed at length in section 8. As presented earlier, the climate ageing results obtained differ from each other to a very small extent and these differences are within the measurement error (of about 20%). And as in the above statement we would like to reiterate that the results of this study are more qualitative than quantitative.

  1. “this sample was much better crosslinked at the forming stage than the other samples because the exothermic peak”. Why and how epoxy resin with 15 phr cellulose had the best crosslinked state? Since all samples were not completely crosslinked, can the author provide the degree of cross-linking of each sample? If not, does it mean that the authors evaluated and compared the performance of the samples under different conditions?

After the reviewer drew attention to this issue, we decided to re-examine the findings of the DSC analysis. We would like to sincerely apologise for our overinterpretation of the results obtained, and the issue in question has been modified accordingly in the manuscript.

At this point we would also like to point out that the study of cross-linking kinetics takes place under different conditions. The study we carried out only aimed to investigate the temperatures of the phase transitions occurring in the material studied.

  1. At the time of reaserch we did not tested degree of cross-linking of the samples. Regardless, all samples were prepared and treated in exactly same conditions. With that said, the crosslinking peak that could be atributed to crosslinking of the resin in tested sample (with 15 phr content of cellulose) may be coused by factor outside our control.”
    • reaserch”, “atribute”, “coused”? While these responses will not appear in your publication, the authors should take it seriously!

We are very sorry for the spelling errors that appeared in our reply.

  • Please answer the question “Why and how epoxy resin with 15 phr cellulose had the best crosslinked state?”
  • What is the “factor outside our control”?

 After the reviewer's comments drew our attention to the results and conclusions we obtained, we decided to reinterpret them. The DSC study we carried out was only to determine the temperatures of phase transitions occurring in the samples studied. The study of crosslinking kinetics takes place under different conditions and the conclusions presented by us turned out to be an overinterpretation of the results obtained. We would like to withdraw from this statement. The manuscript has been modified accordingly.

  1. Please provide TGA results of aged composites, therefore, the influence of ageing processes on the thermal stability of epoxy resin with different amount celluloses can be evaluated.
  • This manuscript investigated “Accelerated ageing of epoxy biocomposites filled with cellulose”, in my opinion, it is necessary to do this test to investigate the influence of ageing processes on the thermal stability of epoxy resin.
  • “no longer have these samples” is not a reason in this situation. What you did is a repeatable scientific work, not a one-shot deal.

  1. The dispersion of cellulose or the uniformity of EP/cellulose composite plays an important role in the analysis of TGA, DSC, and FTIR results. If the composites fabricated in this work are not homogeneous, the credibility of the analysis results (TGA, DSC, and FTIR) of this article will be greatly reduced. If the authors can not provide any results about it, frankly speaking, this work may not meet scientific requirements about this journal.

The methodology and results described below are included in the manuscript. The particle dispersion test was performed using an OPTA-TEC LAB-40 optical microscope and Capture V2.0 software. The test was performed by counting cellulose particles observed in the sample. The magnification of the microscope used to observe the samples was 100 times. The sample area covered by the microscope observation was 12538.78 µm2. Observations were made at 3 different locations of the samples.

“Dispersion analysis was performed for composite samples containing 5, 10, 15 and 20 phr of cellulose. The area tested was 12538.78 µm2. In order to perform the analysis, the number of observed particles was counted from 3 randomly selected parts of the sample. The results of the conducted analysis are presented in the form of a bar chart in Figure 9. Figure 10 presents photographs of the examined areas.

From the collected results it can be observed that the sample containing 15 phr of cellulose obtained the good dispersion degree of 50.33±13.91 particles in the area of 12538.78 µm2. The dispersion of cellulose in the resin at a content of 5 to 10 phr is at a comparable level. We observe weak interactions between the hydrophilic cellulose and the hydrophobic resin, directly translating into a deteriorated dispersion of the filler itself. Merely drying the cellulose is not sufficiently thermally modified to obtain adequate resin reinforcement. Ultimately, part of the resin should be reduced and replaced in part by cellulose fibers in order to maintain the functional properties of the composite, and at the same time have a positive impact on the natural environment. The performed optical tests allow only to determine the general degree of dispersion and distribution of cellulose in the resin.”

Figure 8. Graph showing the number of observed cellulose particles in the composites tested

Cellulose content

Photograph of sample

5 phr

10 phr

15 phr

20 phr

Figure 10. Microscopic images comparing the degree of cellulose dispersion in the samples of composites with 5, 10, 1

Round 3

Reviewer 1 Report

Accept in present form